# Nutritional Intervention with Dried Bonito Broth for the Amelioration of Aggressive Behaviors in Children with Prenatal Exposure to Dioxins in Vietnam: A Pilot Study

**DOI:** 10.3390/nu13051455

**Published:** 2021-04-25

**Authors:** Muneko Nishijo, Tai The Pham, Ngoc Thao Pham, Hai Thai Thu Duong, Ngoc Nghi Tran, Takashi Kondoh, Yoshikazu Nishino, Hiroshi Nishimaru, Quyet Ba Do, Hisao Nishijo

**Affiliations:** 1Department of Public Health, Kanazawa Medical University, Uchinada, Ishikawa 920-0293, Japan; ni-koei@kanazawa-med.ac.jp (M.N.); ynishino@kanazawa-med.ac.jp (Y.N.); 2Biomedical and Pharmaceutical Research Center, Vietnamese Military Medical University, Hanoi 193824, Vietnam; taithuy@kanazawa-med.ac.jp (T.T.P.); phamngocthaovmmu@gmail.com (N.T.P.); dobaquyet@yahoo.com (Q.B.D.); 3Thanh Khe Hospital, Da Nang 50300, Vietnam; bsthuhai@gmail.com; 4Ministry of Health, Vietnam Government, Hanoi 111000, Vietnam; nghi_tranngoc@yahoo.com; 5Department of Food Science and Nutrition, Faculty of Agriculture, Kindai University, Nara 631-8505, Japan; tkon@nara.kindai.ac.jp; 6System Emotional Science, Faculty of Medicine, University of Toyama, Toyama 930-0194, Japan; nishimar@med.u-toyama.ac.jp

**Keywords:** dioxin, inflammation, nutritional intervention, aggression, dried bonito broth

## Abstract

Dioxins have been suggested to induce inflammation in the intestine and brain and to induce neurodevelopmental disorders such as autism spectrum disorder (ASD) and attention deficit hyperactivity disorder (ADHD), partly due to deficits in parvalbumin-positive neurons in the brain that are sensitive to inflammatory stress. Previously, we reported ADHD traits with increased aggressiveness in children with prenatal exposure to dioxins in Vietnam, whereas dried bonito broth (DBB) has been reported to suppress inflammation and inhibit aggressive behavior in animal and human studies. In the present study, we investigated the association between dioxin exposure and the prevalence of children with highly aggressive behaviors (Study 1), as well as the effects of DBB on the prevalence of children with highly aggressive behaviors (Study 2). Methods: In Study 1, we investigated the effects of dioxin exposure on the prevalence of children with high aggression scores, which were assessed using the Children’s Scale of Hostility and Aggression: Reactive/Proactive (C-SHARP) in dioxin-contaminated areas. The data were analyzed using a logistic regression model after adjusting for confounding factors. In Study 2, we performed nutritional intervention by administering DBB for 60 days to ameliorate the aggressiveness of children with high scores on the C-SHARP aggression scale. The effects of DBB were assessed by comparing the prevalence of children with high C-SHARP scores between the pre- and post-intervention examinations. Results: In Study 1, only the prevalence of children with high covert aggression was significantly increased with an increase in dioxin exposure. In Study 2, in the full ingestion (>80% of goal ingestion volume) group, the prevalence of children with high covert aggression associated with dioxin exposure was significantly lower in the post-ingestion examination compared with in the pre-ingestion examination. However, in other ingestion (<20% and 20–79%) groups and a reference (no intervention) group, no difference in the prevalence of children with high covert aggression was found between the examinations before and after the same experimental period. Conclusions: The findings suggest that DBB ingestion may ameliorate children’s aggressive behavior, which is associated with perinatal dioxin exposure.

## 1. Introduction

Environmental pollutants, such as dioxins, induce chronic inflammation in various organs, including the uterus, vessels, intestines, and brain, partly through the aryl hydrocarbon receptor (AhR) [1,2,3,4,5]. Inflammation in the intestine is reported to disrupt the gut microbiota, which may induce neurodevelopmental disorders, including autism spectrum disorder (ASD) and attention deficit hyperactivity disorder (ADHD) [5,6]. Inflammation in the brain may also induce ASD and ADHD [7,8]. Moreover, maternal exposure to environmental toxicants, including dioxin-like polychlorinated biphenyls (PCBs), altered gut microbiota composition, and metabolites in 1-month-old infants, may alter immune system maturation and responses as well as neurodevelopment [9]. These findings suggest that environmental dioxins may induce ASD and ADHD through inflammation in the brain and intestines.

Perinatal dioxin exposure originating from the airbase can influence specific developmental aspects, as shown by the ASD-type difficulties in social interactions as well as poor general neurodevelopment, including language and motor skills, in children from the Da Nang birth cohort recruited in the period 2008–2009 in Da Nang city in Vietnam [10,11,12,13,14]. In these children, particularly in girls, at 3 years of age perinatal dioxin exposure influenced their eating behavior, such as enjoyment of foods, suggesting that the effects of dioxins on emotional development may lead to emaciation/obesity in the future [15]. We also investigated their neurodevelopment at 5 years of age and reported that lower planning ability and poor coordination and motor skills were associated with perinatal dioxin exposure [16].

When children of the Da Nang cohort reached 8 years of age in 2016–2017, a follow-up study was conducted to investigate the effects of prenatal dioxin exposure on the behaviors of children using the ADHD rating scale, including two subscales: inattention and impulsivity-hyperactivity. In girls, significantly increased impulsivity-hyperactivity scores were associated with dioxin exposure, particularly 2,3,7,8-tetrachlorodibenzo-P-dioxin (TCDD), suggesting that TCDD exposure may increase ADHD traits in school-age children in a hot spot of dioxin contamination in Vietnam [17]. As aggressive behavior is often observed among children with ADHD, aggressive behavior among participants in the survey of 8-year-old children was assessed using the Children’s Scale of Hostility and Aggression: Reactive/Proactive (C-SHARP) [17]. This study indicated that covert aggression scores, one of the subscales, significantly increased with increasing TCDD exposure in girls [17].

Dried bonito broth (DBB), a traditional Japanese food containing abundant histidine [18], has been reported to reduce the blood levels of inflammatory cytokines (interleukin (IL)-5, -6, and -13) in humans [19]. Moreover, in our previous animal study, the ingestion of DBB decreased aggressiveness, increased latency, and decreased the number of attacks of resident mice when an intruder was introduced into the home cage [20]. In the present study, we hypothesized that DBB ingestion may ameliorate aggressiveness in children exposed to dioxins that have been reported to induce chronic inflammation and neurodevelopmental disorders (see above for detail). Therefore, we first investigated the associations between perinatal dioxin exposure, the levels of which are reflected by dioxins in their maternal breast milk, and the prevalence of children at risk of aggressive behavior in the 8-year-old survey in a hot spot of dioxin contamination in Vietnam. Second, we investigated the possible beneficial effects of DBB ingestion to ameliorate aggressive behavior associated with dioxin exposure, particularly TCDD exposure, in children during the perinatal period.

## 2. Materials and Methods

The present study comprised two parts: Study 1 investigated the association between perinatal dioxin exposure, particularly TCDD exposure, and the prevalence of children with high aggression scores, and Study 2 investigated the effects of DBB ingestion on aggressive behaviors in children found to be at risk in Study 1.

### 2.1. Study 1

#### 2.1.1. Subjects in Study 1

The Thanh Khe and Son Tra districts in Da Nang city were chosen as our study areas. These districts are located within 10 km of the former Da Nang airbase and are hot spots of dioxin contamination originating from herbicide spraying in Vietnam. The subjects were children from the Da Nang birth cohort, including 238 mother and infant pairs (158 pairs in Thanh Khe and 80 pairs in Son Tra) recruited in the period 2008–2009; we reported the impacts of dioxins on their growth and neurodevelopment in follow-up studies from 1 month later to the present time [10,11,12,13,14,15,16,17].

When the children of the Da Nang cohort reached 8 years of age in 2016–2017, a follow-up survey (Study 1) was conducted to investigate their behavior, including aggressive behavior. A total of 181 children (114 in Thanh Khe district and 67 children in Son Tra district) participated in the survey. Table 1 shows the characteristics of the children’s mothers and families, the children at birth and in the survey, and the perinatal dioxin exposure levels indicated by TCDD and the toxic equivalency values of polychlorinated dibenzodioxins and polychlorinated dibenzofurans (TEQ-PCDD/Fs) in maternal breast milk. In both Thanh Khe and Son Tra, no significant differences between groups were observed for almost all factors, except for the rates of primiparae in the Son Tra area.

The study design was approved by the institutional ethics board for medical studies at Kanazawa Medical University (No. 16–19, 26 September 2016). Written informed consent was obtained from all mothers according to the procedures reviewed and approved by the Health Department of Da Nang City and the Vietnamese Military Medical University.

#### 2.1.2. Behavior Assessment: C-SHARP Aggression Scale

Aggressive behavior was examined by interviewing the parents or caregivers of children using the C-SHARP aggression scale developed by Farmer and Aman (2009) [21] with 5 subscales (verbal aggression, bullying, covert aggression, hostility, physical aggression). The same examiner, who was blind to the dioxin exposure levels and developmental status of the children, interviewed in each area in Study 1 as well as in Study 2.

Children at risk of aggressive behaviors were estimated based on subscale scores higher than cut-off values defined as means + standard deviations (SDs) of the subscale scores in the general population reported by Farmer and Aman (2010) [22]. The cut-off values for verbal aggression, bullying, covert aggression, hostility, and physical aggression were 8.1, 10.1, 10.1, 12.1, and 4.1 points, respectively.

#### 2.1.3. Exposure Assessment and Data Analysis

The perinatal dioxin exposure of children was estimated using the dioxin levels in their mothers’ breast milk collected one month after giving birth, since a previous study reported significant correlations between dioxin levels in breast milk and those in cord blood [23]. Seventeen 2,3,7,8-substituted congeners of polychlorodibenzodioxins (PCDDs) and polychlorodibenzofurans (PCDFs), including 2,3,7,8-tetrachlorodibenzo-*p*-dioxin (TCDD), were measured using a gas chromatograph (HP-6980; Hewlett-Packard, Palo Alto, CA, USA) equipped with a high-resolution mass spectrometer (high-resolution-gas chromatography/mass spectrometry; MStation-JMS700, JEOL, Tokyo, Japan). More details of dioxin analysis were reported in our previous study [24]. The calculations of TEQ-PCDDs/PCDFs were referenced from the WHO 2005 toxic equivalent factor [25].

Increased risks (odds ratios) for high C-SARP aggression subscale scores associated with an increase in TCDD exposure were analyzed after adjusting for confounding factors, including the education of mothers, family income, area, sex, and the child’s age at the survey, using binary logistic regression analysis.

### 2.2. Study 2

#### 2.2.1. Subjects in Study 2

In at least one of the C-SHARP subscale scores, 41 children (36.0%) in Thanh Khe and 23 children (34.3%) in Son Tra scored higher than the mean + standard deviation (SD) score of the general population [21,22]. Forty children who scored higher on at least one subscale and had a refrigerator at home in Thanh Khe were assigned to the intervention group of DBB ingestion, and all of them participated in the post-intervention examination using C-SHARP. The 23 children in Son Tra with higher aggression scores were assigned to the control group, but a C-SHARP score for one child could not be obtained in the post-intervention survey because of his/her caregiver’s absence. Therefore, 40 children in Thanh Khe (DBB-intervention area) and 22 children in Son Tra (reference area) were included in the final analysis. The characteristics of the children who participated in Study 2 in each area are shown in Table 1.

Written informed consent was obtained from all of the mothers according to the procedures reviewed and approved by the Health Department of Danang City and the Vietnam Military Medical University. The institutional ethics board for medical studies at Kanazawa Medical University (No. 16–19, 26 September 2016), University of Toyama (No. 28–82, 17 October 2016), and Ajinomoto Co., Inc. (No. 2016-019, 16 September 2016), approved the study design and all experimental procedures.

#### 2.2.2. Intervention Methods

In the DBB-intervention area, the children were provided with DBB. DBB (Hondzukuri Ichiban-Dashi Katsuo; Ajinomoto Co., Inc., Tokyo, Japan) extracted from traditional dried bonito shavings was used for the intervention. The macronutrient composition (per 100 g dashi) includes 3.44 g protein, 0.8 g ash, 127 mg sodium, and 596.1 mg histidine [18]. DBB was divided into 50 mL plastic tubes for the ingestion of 30 mL at a time and stored in freezers at home. Mothers and/or caretakers prepared the DBB for the children to take a total of 30 mL of DBB with regular meals every day for 60 days. They also completed a DBB intake diary to estimate on how many days and how much DBB the children ingested during the intervention.

In the reference area, the parents were provided with advice for the education of the children after the pre-intervention examination of C-SHARP in Study 1.

#### 2.2.3. Post-Intervention Examination in Than Khe (DBB-Intervention Area)

After 60 days of the intervention, the children in Thanh Khe (DBB-intervention group) underwent a post-intervention examination using C-SHARP. The prevalence of children with high aggression scores in each subscale was compared between the pre-intervention and post-intervention examinations using the McNemar test.

#### 2.2.4. Post-Intervention Examination in Son Tra (Reference Area)

Two months (60 days) after the pre-intervention examination, to assess the effects of the intervention (educational advice), the children in Son Tra (reference group) were re-examined using C-SHARP. The prevalence of children with high aggression scores in each subscale in the survey of Study 1 (pre-intervention examination) was compared with that in the post-intervention examination in Study 2 using the McNemar test.

### 2.3. Statistical Analysis

SPSS version 21.0 (IBM SPSS, Armonk, NY, USA) was used for all statistical analyses. The level of significance was set at *p* < 0.05.

## 3. Results

### 3.1. Associations between the Prevalence of Children with High Aggression Scores and Dioxin Exposure (Study 1)

Table 2 shows the adjusted odds ratios for the risk of children showing high C-SHARP aggression subscale scores due to increased TCDD exposure. The prevalence of children with high covert aggression subscale scores was significantly increased with an increase in TCDD exposure (4.49 of adjusted odds ratio), but other aggression subscale scores were not. Although we also investigated the association between the prevalence of children with high aggression subscale scores and TEQ-PCDD/Fs exposure, no significantly increased risk with an increase in TEQ-PCDD/Fs was observed (data not shown).

### 3.2. Effects of DBB Ingestion on Aggression Scores in Than Khe (Study 2)

Based on the records in the DBB intake diary, the percent ingestion, defined as the actual ingestion volume (mL) relative to the goal volume (1800 mL), was estimated for each child in Thanh Khe. The distribution of the percent ingestion of DBB shown in Table 3 indicates three peaks, including 9 children at the 0–9.9% level, 6 children at the 40.0–49.9% level, and 14 children at the 100% level. Additionally, no children were found at the 20–29.9% level between the first and second peaks and at the 70.0–79.9% level between the second and third peaks. Therefore, the children were divided into three groups according to their percent ingestion: no ingestion (0.0–19.9%), partial ingestion (20.0–79.9%), and full ingestion (≥80%). The numbers of the subjects and means with the SDs of the DBB ingestion volume in the three groups are shown in Table 3.

Due to the remarkable differences in DBB ingestion among the DBB-intervention groups, the prevalence of children with high aggression subscale scores in Thanh Khe was compared between the pre- and post-intervention examinations in each group with different levels of DBB ingestion (Table 4). In the no ingestion and partial ingestion groups, no significant differences in the prevalence of children with high scores were found in any subscale. In the full ingestion group, however, the prevalence of children with a high score for covert aggression was significantly decreased in the post-intervention examination compared with in the pre-intervention examination. The prevalence of children with a high score in the bullying subscale was decreased in the post-intervention examination compared with in the pre-intervention examination, but the difference was not significant.

### 3.3. Comparisons of the Prevalence of Aggression Scale Scores between Two Examinations in Son Tra (Reference Group with No DBB Ingestion)

In Son Tra (reference group for intervention), the prevalence of children with high C-SHARP scores in the pre-intervention examination in Study 1 was compared with that in the post-intervention examination in Study 2 (Table 5). The prevalence of children with high hostility and physical aggression subscale scores was significantly lower in the post-intervention examination in Study 2 than in the pre-intervention examination in Study 1. However, there was no significant difference in the prevalence of children with high covert aggression subscale scores.

## 4. Discussion

### 4.1. Effects of DBB Ingestion on C-SHARP Aggression Scores in Vietnamese Children Exposed to Dioxins

Firstly, in Study 1 we investigated the association between dioxin exposure and increases in the prevalence of children with high aggression scores. Previously, Pham-The et al. [17] reported that covert aggression subscale scores increased with increasing TCDD exposure in children from this cohort, but it was significant only in girls. The present study (Study 1) indicated that TCDD exposure was significantly associated with increases in the prevalence of children with high covert aggression subscale scores in all children in the pre-intervention examination after adjusting for confounding factors, including the sex of children. In Study 2, we found that the full ingestion of DBB decreased the prevalence of children with high covert aggression subscale scores, while partial and no ingestion did not affect this prevalence in children in Than Khe (DBB-intervention area). Furthermore, the same period of 60 days without DBB ingestion also did not affect the prevalence of children with high covert aggression subscale scores in Son Tra (reference area). These results suggest that a DBB ingestion of more than 1440 mL (80% of 1800 mL) may ameliorate covert aggression behavior associated with perinatal TCDD exposure in children living in a hot spot of dioxin contamination in Vietnam.

Unexpectedly, in Thanh Khe half of the children gave up DBB ingestion before reaching a sufficient volume for successful intervention. In particular, 12 children took only 91 mL of mean DBB volume (<20%), indicating almost no ingestion (i.e., no ingestion group). In this group, however, the prevalence of children with high aggression subscale scores in the pre-intervention examination was lower than that in the pre-intervention examination in the partial and full ingestion groups (Table 4). It is possible that the learning ability in the no ingestion group might be higher than that in the partial and full ingestion groups, and, consequently, these children might have shown better behavior at home and school. This may explain why their parents allowed the children to stop DBB ingestion.

In Son Tra (reference area), the prevalence of children with high hostility and physical aggression subscale scores was significantly decreased after 60 days without DBB ingestion. Since increases in these aggression subscale scores were not associated with dioxin exposure, our advice to parents for the education of children might improve relationships between parents and children, which could thus lead to decreased hostility and physical aggression subscale scores.

### 4.2. Nutritional Intervention to Improve Behaviors in Children with ADHD

To our knowledge, no intervention using DBB ingestion has been performed in children with highly aggressive behaviors before this. However, nutritional intervention with supplements of multi-nutrients, including vitamins and minerals, and polyunsaturated fatty acids, has been investigated in children with ADHD [26,27], who often show aggressive behavior, and has been reported to improve behavior. The children with high exposure to dioxins in the present study had not only high covert aggression subscale scores but also high hyperactivity scores [17], suggesting that DBB might influence behaviors related to ADHD symptoms. In the future, we will conduct an intervention study using DBB to investigate the effects of DBB on ADHD scores, ADHD symptoms, and aggression subscales.

### 4.3. Possible Mechanisms of DBB Effects

In our previous study, exposure to TCDD during the fetal period decreased the number of parvalbumin (PV)-positive neurons in the basolateral amygdala, hippocampus, and superior colliculus in both sexes of offspring rats [28], suggesting that TCDD exposure influences neuronal development in the brainstem and limbic system, particularly the amygdala and hippocampus. PV-positive neurons are a subclass of GABAergic inhibitory interneurons that control the outputs of pyramidal neurons and facilitate sensory and cognitive information processing through gamma oscillation [29,30,31,32]. Chronic inflammation induces oxidative stress [33], and as PV-positive neurons mature slowly, oxidative stress might induce PV-positive neuronal loss and/or disturb their maturation [34,35,36]. Furthermore, ASD and ADHD symptoms may be associated with alterations in PV-positive neuronal functions [37,38]. These findings suggest that dioxins may induce increases in aggressiveness through their adverse effects on PV-positive neurons. However, DBB has anti-inflammatory effects (see Introduction). Furthermore, DBB includes substances such as histidine, anserine, and phenolic compounds, which have been reported to show antioxidative actions in vitro [39,40,41]. Furthermore, DBB decreases the urinary biomarker of oxidative stress in humans [42]. Consistently, DBB intake decreased aggressiveness, which was associated with increased densities of PV-positive neurons in the medial prefrontal cortex, amygdala, and hippocampus in mice [20]. Taken together, decreased aggression after an intervention with DBB may be attributed to the increase in the number or activity of PV-positive neurons in the brains of children.

DBB contains plenty of histidine [18], which is a precursor of histamine, which is a trophic and/or neurogenic factor in the developing brain [43]. In 3-year-old children from the current Da Nang cohort, lower urinary histidine levels were reported in children with high TCDD exposure and poor neurodevelopment compared with non-exposed and developmentally normal children [44], suggesting lower levels of histamine in the brains of children with high TCDD exposure. These results suggest that DBB ingestion might increase the histamine levels in the brain and alter behavior in children with high TCDD exposure. Histamine was reported to directly open GABA_A_ receptors and/or potentiate the current mediated by GABA_A_ receptors in vitro [45], as well as exciting GABAergic neurons in the septohippocampal system [46]. These facts suggest that DBB ingestion might facilitate PV-positive neuronal functions, leading to the inhibition of aggressive behavior.

### 4.4. Limitations

There were several limitations in the present study. The first limitation is the small number of children, with only 20 children who ingested more than 80% of the goal volume of DBB. Eating the same food every day for 2 months is not easy for young children. However, the present results suggest that ingestion for nearly 2 months is necessary to affect behavior in children. Furthermore, there were some differences in the prevalence of children with high C-SARP aggression subscale scores in the pre-intervention examination among the three DBB-intervention groups (Table 4), which might be ascribed to the small number of children in the analysis. Therefore, we compared the data between the pre- and post-intervention examinations within each group. The second limitation is the method—parent rating scales—that we used to evaluate the effects of DBB ingestion. The same interviewer asked the questions in a similar way, but repeated the same questions twice, which might have influenced the participants’ answers, particularly in children without DBB intervention in Son Tra. The final limitation is that the DBB we used in the present study (Hondzukuri Ichiban-Dashi Katsuo; Ajinomoto Co., Inc., Tokyo, Japan) is an extraction (liquid) that is not easy to obtain in Vietnam because it is not customary to use DBB for cooking. To recommend the ingestion of DBB to Vietnamese children in dioxin-exposed areas, other forms of DBB, such as freeze-dried DBB, should be developed in the future.

## 5. Conclusions

DBB ingestion decreased the prevalence of children with high covert aggression associated with perinatal dioxin exposure in a hot spot of dioxin contamination in Vietnam, suggesting the possible beneficial effects of DBB to ameliorate aggressive behaviors in children exposed to dioxins.

## Figures and Tables

**Table 1 nutrients-13-01455-t001:** Characteristics and dioxin exposure levels of all children who participated in the pre-intervention survey (Study 1) and those of children who were assigned to the intervention study in Thanh Khe (DBB-intervention area) and Son Tra (reference area) (Study 2).

		All Children (Study 1)				Children Selected for Study 2			
Area		Thanh Khe		Son Tra		Thanh Khe		Son Tra	
*n*		114		67		40		22	
		Mean (*n*)	SD (%)	Mean (*n*)	SD (%)	Mean (*n*)	SD (%)	Mean (*n*)	SD (%)
Mother and Family	Maternal age (years)	28.7	6.2	27.7	5.9	28.4	6.7	30.6	5.9
	Maternal education (years)	8.5	3.5	8.7	3.5	8.9	3.4	9.2	3.4
	Parity (primiparae)	(31)	(27.2)	(19)	(28.4)	(14)	(35.0)	(2)	(9.1)
	Family smoking	(95)	(83.3)	(55)	(82.1)	(31)	(77.5)	(18)	(81.8)
	Maternal drinking habit	(14)	(12.3)	(16)	(23.9)	(6)	(15.0)	(5)	(22.7)
	Family income (million VND)	2.8	1.5	3.3	1.9	2.8	1.5	3.4	2.1
Children	Sex (boys)	(60)	(52.6)	(45)	(67.2)	(17)	(42.5)	(9)	(40.9)
	Gestational weeks	39.6	0.8	39.6	0.8	39.6	0.7	39.6	0.5
	Birth weight (g)	3222	395	3255	396	3257	292	3207	387
	Age at the survey (years)	7.8	0.15	7.7	0.04	7.8	0.14	7.7	0.04
	Weight at the survey (kg)	25.8	7.1	27.6	6.4	25.6	6.1	28.3	5.2
	Height at the survey (cm)	123.9	5.2	125.3	4.7	123.7	4.9	126	4.2
	BMI at the survey	16.7	3.7	17.4	3.2	16.6	3.1	17.7	2.8
Dioxins in Breast Milk	TCDD (pg/g lipid)	1.7	2.1	1.1	2.4	1.8	2.2	1.2	2.6
	TEQ-PCDD/Fs (TEQ/g lipid)	13.7	1.6	11.6	1.6	13.5	1.7	11.9	1.6

*n*: number of subjects; SD: standard deviation; VND: Viet Nam Dong; TCDD: 2,3,7,8-tetrachlorodibenzo-p-dioxin; TEQ-PCDD/Fs: toxic equivalency-polychlorinated dibenzo-p-dioxins and polychlorinated dibenzo furans.

**Table 2 nutrients-13-01455-t002:** Increased risks of high C-SARP subscale scores associated with an increase in TCDD exposure.

C-SHARP Subscale	*n* (%)	OR for an Increase in TCDD	95% CI		*p*-Value
Verbal	14 (7.7)	0.87	0.14	5.25	NS
Bullying	17 (9.4)	0.71	0.15	3.36	NS
Covert	32 (17.6)	4.49	1.06	19.0	0.041
Hostility	29 (16.0)	3.52	0.92	13.5	NS
Physical	25 (13.8)	1.01	0.24	4.23	NS

*n*: number of subjects; SD: standard deviation; OR: odds ratio, CI: confidence interval. Covariates: education of mothers, family income, area, sex, child age at the survey. Cut-off values for high scores were 8.1, 10.1, 10.1, 12.1, and 4.1 points for verbal, bullying, covert, hostility, and physical aggression subscales, respectively, according to Farmer and Aman (2010) [22].

**Table 3 nutrients-13-01455-t003:** DBB ingestion volume and intervention categories in children in Thanh Khe (DBB-intervention area).

Ingestion (%)	*n*	Ingestion Group	Number of Subjects in Each Group	Mean of DBB Intake Volume (mL)	SD of Intake Volume
0.0−9.9	9	No	12	19	10.7
10.0−19.9	3				
20.0−29.9	0	Partial	8	780	159
30.0−39.9	1				
40.0−49.9	6				
50.0−59.9	0				
60.0−69.9	1				
70.0−79.9	0				
80.0−89.9	3	Full	20	1744	112
90.0−99.0	3				
100	14				

Ingestion (%): actual ingestion volume (mL)/1800 mL (ingestion goal); *n*: number of subjects; SD: standard deviation.

**Table 4 nutrients-13-01455-t004:** Comparisons of the prevalence of children with high C-SARP aggression subscale scores between the pre- and post-intervention examinations in the DBB-intervention group in Thanh Khe.

Ingestion Group	No (*n* = 12)					Partial (*n* = 8)					Full (*n* = 20)				
Examination	Pre		Post		*p*-Value	Pre		Post		*p*-Value	Pre		Post		*p*-Value
Subscales	*n*	%	*n*	%		*n*	%	*n*	%		*n*	%	*n*	%	
Verbal	0	0	0	0	NS	3	38	1	13	NS	3	15	1	5	NS
Bullying	0	0	0	0	NS	3	38	3	38	NS	7	35	1	5	NS
Covert	4	33	4	33	NS	5	63	4	50	NS	14	70	7	35	0.039
Hostility	1	8	0	0	NS	4	50	2	25	NS	10	50	4	20	NS
Physical	2	17	2	17	NS	2	25	1	13	NS	2	10	3	15	NS

*n*: number of subjects; *p*-value for comparison between the pre- and post-intervention examinations by McNemar test; NS: not significant.

**Table 5 nutrients-13-01455-t005:** Comparisons of the prevalence of children with high C-SARP aggression subscale scores between Study 1 and Study 2 in the reference group in Son Tra.

Examination	Pre-Intervention Examination (Study 1)		Post-Intervention Examination (Study 2)		*p*-Value
Subscales	*n*	%	*n*	%	
Verbal	5	23	0	0	NS
Bullying	5	23	0	0	NS
Covert	4	18	4	18	NS
Hostility	12	55	3	14	0.012
Physical	9	41	1	5	0.021

*n*: number of the subjects; *p*-value for comparison between the pre- and post-intervention examinations by McNemar test; NS: not significant.

## Data Availability

The data presented in this study are available on request from the corresponding author.

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
