# Peer review of "Nutritional Intervention with Dried Bonito Broth for the Amelioration of Aggressive Behaviors in Children with Prenatal Exposure to Dioxins in Vietnam: A Pilot Study"

_nutrients, 2021, doi:10.3390/nu13051455_

Round 1
Reviewer 1 Report
The manuscript was adressed acording to the first round of Reviw and now it was strongly improved. Especially, the different subheading between Trial 1 and 2 has been very enlightening.
With a only miror point, in Tables the number of samples was cited as capital N (N). Usually, the number of samples investigated is often described as "n". Thus, I suggest to change to the most usual form to avoid confusion on the part of the reader
Reviewer 2 Report
- The authors tried to sincerely respond to this reviewer’s comments. However, unfortunately, I did not understand what the authors tried to mention because of too many English language problems: not only simple mistakes but also constructions. The authors must have the entire text checked by a professional English language service. For example, in Discussion (L301 around), the sentence reads, “The present study indicated that TCDD exposure was significantly associated with increases in the prevalence of children with DBB ingestion in Thanh Khe, but not high covert aggression score in all children after adjusting confounding factors including sex of children”. This sentence is too complicated to convey what the authors probably want to say. TCDD exposure must have some relationship with the prevalence of covert aggression in children who ingested specific amounts of DBB, but not with the prevalence of children with DBB. It was not clear how TCDD exposure is related to high covert aggression score? The last half of this sentence does not make sense at all. I would think that the sentence should be split into at least two. I am afraid that there are many similar problems throughout the text.
- Regarding minor editing problems, the authors may find it helpful to use computer-assisted English writing apps, such as Grammarly. For example, L156-L161: Remove “a” from “Children at a risk”; “bulling” should be “bullying”; Insert “the” before “general population”
- Abstract: This study's aims should be described in the end of introductory remarks, not in “Methods”. Specific details of the methods are missing here and should be concisely described. L.19-L.21: This statement is too strong to be supported by the previous studies.
- L.190 What does it mean by a phrase, “a process that had been reviewed”?
- L.197, The newly added sentence seems redundant and should be removed.
- L.207-L.208. I do not understand what the authors meant to say here. What is the relationship of education advice with the provision of DBB?
- L.222. What is the relationship of the advice with the second examination?
- L.232. I cannot understand what the authors meant to say.
Round 2
Reviewer 2 Report
The quality of the manuscript has significantly improved. However, there are still many wording problems. I am afraid that this manuscript is still at a preparative stage. Because the authors may not understand what problems this reviewer raised, they are requested to study the following paragraph as an example of inappropriate wording.
L293-L299: In this group, however, the prevalence of children with high aggression subscale scores in the pre-intervention examination was lower than those in the pre-intervention examination in the partial and full ingestion groups (Table 4). It is possible that the learning ability in the no ingestion group might be higher than those in the partial and full ingestion groups, and consequently, they might show better behavior at home and school. This may explain why their parents allowed children to stop DBB ingestion.
Problems:
1) The prevalence should be followed by signs and symptoms, or disease, rather than “children.” You may find “Google Scholar” helpful to check the usage. This expression appears many times in the text. 2) What does “those” represent?
3)A phrase like “It is possible that” is wordy and should be replaced with more appropriate wording. 4) “full” and “ingestion” should be described as “full-ingestion” because you do not mean full groups. 5) Who are “they”? 6) It may be unclear what “This” refers to.
Throughout the text, there are many similar problems, and this reviewer would strongly suggest again the authors ask a professional Language service, not for simple grammatical problems but advanced writing editing.
Author Response
Response: We thank the comment very much. According to the comment, we applied to the English Editing Service by MDPI. All sentences of the manuscript have been corrected by this service. The changes are indicated by the track changes function of MS Word.
This manuscript is a resubmission of an earlier submission. The following is a list of the peer review reports and author responses from that submission.
Round 1
Reviewer 1 Report
General comments:
The work is based on a frankly ingenious and unusual hypothesis, such as subjecting children with an average age of slightly less than 8 years to a diet with the same food for two months. From the nutritional point of view, it is certainly not the most advisable option.
Despite its originality, the work has some design problems that in my opinion make the results obtained at least, of doubtful reliability.
Firstly, it gives the impression that the only source of dioxin intake by these children is breast milk, which is not true or at least inaccurate. There are other food sources through which dioxins can reach children, which have not been duly quantified in this work.
Furthermore, the relevant results of the work are based only on qualitative results freely observed by the medical team, which is always more subjective than the quantitative parameters. The results obtained do not achieve great statistical differences either. For example, in Table 3, a result with P = 0.07 (which according to those established in the materials and methods should not have been considered significant) and another one that reaches a level of P = 0.039 are shown as statistically significant differences. In this case, a single misclassified observation would also distort the result, making it not statistically significant.
Overall, I consider that the work has more than relevant weaknesses to consider that the conclusions presented, as well as the title of the work, are not sufficiently demonstrated by the results obtained
Minor comments:
Abstract section: C-SHARP was not defined previously to be cited in the text.
Page 2, lines 51-53 “Moreover,….infants” This phrase seems to be incomplete.
Page 2, lines 68-81: This paragraph is not introduction. It is a combined paragraph that included materials and methods and even results.
Reviewer 2 Report
This paper reported the possible beneficial nutritional effect of bonito extract broth on the amelioration of aggressive behavior in dioxin exposed children in Viet Nam's hot spot area. This study's aim is challenging and provocative to a readership in a variety of research fields in biomedical sciences. Although the paper is well written in English, it may lack the clarity of methodology and seem ambiguous in interpreting the investigation results. I describe the major and minor points below. I also highlighted the sentences or words in the text in the attached file for reference purposes.
Major:
- Table 3. The authors should clearly describe how they defined "high" verbal, bullying, covert, hostility, and physical score. It seems appropriate to delete "high" when the authors indicate the parameters they used.
- Table 3. The authors divided the group of the Thanh Khe area into three groups, No, Partial, and Full. How did they define the cutoff values between the groups? Is there any possibility to obtain results that may change the conclusion if different cutoff values are used?
- Was the behavioral examination performed on the blind-basis?
- Table 3. The authors compared the possible differences in a particular behavioral score between pre-intervention and post-intervention. Why is the percentage of a particular behavioral score in pre-intervention in "No," "Partial," and "Full" likely to be different from each other?
- The authors cited their work (Ref. 17), which is only a conference abstract, stating that covert aggression scores, one of the subscales, significantly increased with increasing TCDD in girls. I would recommend that the authors strengthen the present study results by incorporating such a dose-response relationship in this paper. Also, Supplementary Table 1 should be incorporated into the text.
Minor:
- Lines 166_172. Because "pre-examination" and "post-intervention" used in the respective study areas are described, it is hard to comprehend the study design. I would suggest that the authors should separately write a survey of each study area.
- I have highlighted several places in the text, which is grammatically correct but is misleading or too ambiguous for readers to understand. For example, L. 30_L33. This sentence could be more appropriately written as follows. "We performed nutritional intervention by administering DBB for 60 days to ameliorate the aggressiveness of children with high scores on the C-SHARP aggression scale in dioxin-contaminated areas. In Table 1, the authors are suggested to reexamine the highlighted numbers in terms of the effective figures. I would suggest that the authors should carefully reexamine the rest.
